# Signature of cardiac alterations in early and late chronic infections with *Trypanosoma cruzi* in mice

**Bárbara Carolina Arias-Argáez**[1], **Victor Manuel Dzul-Huchim**[1], **Ana Paulina Haro-Álvarez**[2], **Miguel Enrique Rosado-Vallado**[1], **Liliana Villanueva-Lizama**[1], **Julio Vladimir Cruz-Chan**[1]*, **Eric Dumonteil**[3]*

1 Laboratorio de Parasitología, Centro de Investigaciones Regionales "Dr. Hideyo Noguchi", Universidad Autónoma de Yucatán, Mérida, Yucatán, México, 2 Instituto de Investigaciones en Ciencias Veterinarias, Universidad Autónoma de Baja California, Mexicali, Baja California, México, 3 Department of Tropical Medicine and Infectious Disease, School of Public Health and Tropical Medicine, and Vector-Borne and Infectious Disease Research Center, Tulane University, New Orleans, Louisiana, United States of America

* vladimir.cruz@correo.uady.mx (JVCC); edumonte@tulane.edu (ED)

**Data Availability Statement:** All relevant data are within the paper and its Supporting Information files.

## Abstract

Chagas disease by *Trypanosoma cruzi* (*T. cruzi*) infection is a leading cause of myocarditis worldwide. Chagas cardiomyopathy is presented with a wide variety of conduction abnormalities including arrhythmias, first- and second-degree atrioventricular blockade, left ventricular systolic dysfunction and some cases heart failure leading to the death. Currently, there are no effective treatments available against advanced Chagas disease. With the advance in the development of novel therapies, it is important to utilize an animal model that can effectively replicate the diverse stages of Chagas disease, including chronic asymptomatic and symptomatic infection, that are akin to those observed in humans. Therefore, to characterize the cardiac alterations during the evolution of the infection, we evaluated the progression of cardiomyopathy caused by *T. cruzi* H1 infection in both BALB/c and ICR mouse models by performing electrocardiogram (ECG) studies in unanesthetized mice every month until 210 days post-infection (dpi). In the late chronic phase of infection, we also performed echocardiogram (ECHO) studies to further assess cardiac function. In conclusion, we demonstrated that ICR mice were more susceptible to cardiac alterations compared to BALB/c mice and both mouse strains are suitable experimental models to study chronic *T. cruzi* infection and novel treatments.

## Introduction

Chagas disease is a worldwide health problem caused by the protozoan parasite *Trypanosoma cruzi* (*T. cruzi*) with 6–7 million people infected worldwide, mainly settled in the Americas [1]. The disease initiates with an acute phase which lasts a few weeks and is characterized by parasites in the bloodstream and unspecific symptoms such as nausea, headache, and fever. Next, infected individuals progress into the symptomatic or asymptomatic chronic phase, where the

**Funding:** This work was supported by Fundación Carlos Slim (#87714) and Consejo Nacional de Ciencia y Tecnología, México (CONACYT, #PDCPN2015-102). B.C.A.A was supported by scholarship #334483 from CONACYT. The funders had no role in study design, data collection and analysis, decision to publish, or preparation of the manuscript.

**Competing interests:** The authors have declared that no competing interests exist.

latter is delimited by the absence of abnormalities in the physical examination and cardiac monitoring, while symptomatic individuals, about 30 to 40% of cases, evolve to megaesophagus, megacolon, and cardiomyopathy years after the infection [2, 3].

It is widely acknowledged that the pathogenesis of cardiomyopathy in Chagas disease is predominantly attributed to the persistent presence of *T. cruzi* within cardiac tissue, leading to inflammation and fibrosis. These changes culminate in structural alterations to the cardiac conduction system, resulting in the development of cardiac rhythm abnormalities such as bradyarrhythmia and tachyarrhythmia which in some severe cases, can progress to heart failure [4]. In the early chronic phase infected human patients exhibit abnormalities in myocardial contractility and left ventricular wall motion, which generally are associated with ventricular arrhythmia and cardiac fibrosis areas. [3, 5–7]. Studies in murine models infected with *T. cruzi* have reported differences in cardiac parameters performed by electrocardiogram (ECG) and echocardiogram (ECHO) recordings compared to naive mice. BALB/c mice infected with *T. cruzi* Tulahuen strain developed alterations in the QRS complex as well as prolonged QT intervals in the first 5 weeks of infection, and these alterations were maintained during the early chronic phase at 90 days post-infection (dpi) [8]. Other alterations have been reported in BALB/c mice infected with *T. cruzi* Colombian strain, where mice displayed prolonged P, PR, and QTc wave intervals, as well decreased heart rate. In addition, an increase in the right ventricle area, a reduction in the ejection fraction and elevated systolic volume in the left ventricle were observed by ECHO assays, during the early chronic phase at 90 dpi [9]. The longest study for chronic infection was performed in ICR mice infected with *T. cruzi* Brazilian strain at 150 dpi which showed a reduction in HR, left ventricular systolic function, and right/left ventricular dilation by ECHOs [10].

The chemotherapy of *T. cruzi* infections is based on nitrofurans and nitroimidazoles which are unsatisfactory since both compounds have toxic-side effects, ineffective in the chronic phase, and are associated with drug resistance [11, 12]. An alternative approach is the development of a vaccine, which could be administered as immunotherapy either to individuals with acute or chronic infection to prevent the development of cardiomyopathy [13–18]. Nevertheless, despite efforts, a vaccine is still in preclinical studies and given the difficulties to perform non-human primates studies, there is an urgent need to optimize an adequate animal model that reproduces ECG and ECHO abnormalities which are the signature of Chagas disease cardiomyopathy [19]. Thus, we aimed here to characterize the progression of chronic Chagas disease cardiomyopathy in experimentally infected mice based on ECG and ECHO findings up to 210 dpi.

## Material and methods

### Ethical standards

All experimental protocols were approved by the institutional bioethics committee of the "Centro de Investigaciones Regionales Dr. Hideyo Noguchi", Universidad Autónoma de Yucatán (Reference #CEI-08-2019) and were performed in strict compliance with the Official Mexican Standards (NOM-062-ZOO-1999).

### Mice and parasites

Female BALB/c (n = 13) and ICR (n = 13) mice were obtained at 4–5 weeks old (Envigo, México) and infected with *T. cruzi* H1 strain, originally isolated from a human case in Yucatán-México and maintained by serial passage in BALB/c mice as previously described [20]. Control groups included non-infected BALB/c (n = 3) and ICR (n = 3) mice. All animals were housed on a 12-hour light/dark cycle in groups of 3–5 per cage attached with Smart Flow

ventilation system (Tecniplast) and air change of 75 times per hour. Animals received food and water *ad libitum* by staff with more than 3 years of experience in animal care. Mice health and behavior were monitored daily, and cages were cleaned weekly. In order to minimize animal stress, we provide them of paper and cardboard tubes previously sterilized by ultra-violet light. Survival was recorded up to 210 dpi which was the end of the study. The euthanasia was performed by deep anesthesia induced with xylazine/ketamine (10mg/kg / 100mg/kg) followed by cervical dislocation. Weight loss, poor body skin or fur condition were considered as sign of endpoint criteria.

## Electrocardiography

Electrical cardiac activity was recorded from non-anesthetized infected mice every month until 210 dpi using ECGenie equipment (Mouse Specifics Inc.). Mice were subjected to an adaptation period to minimize stress in the electrocardiograph plate by placing them for 10 min before recordings. The ECGs trace was obtained with standard lead (dipolar lead DII) and amplitude settings to give 2 mV/1 msec. Typically, 20–25 well-defined successive beats were analyzed from each tracing using the EzCG Signal Analysis Software package (Mouse Specifics Inc.). The following ECG-wave parameters were obtained: heart rate (HR), RR interval, PQ interval, PR interval, QT interval, QTc interval, QRS complex, and ST segment.

## Echocardiography

To induce anesthesia in mice, isoflurane was administered at a dosage of 3% along with 0.5 L/min $O_2$ in an anesthetic chamber (Patterson Scientific). The mice were then maintained at a dosage of 1.5–2.5% isoflurane using a face mask. Echocardiography (ECHO) recordings were obtained using a 22 MHz linear-array ultrasound transducer of the Mylab Seven system (Esaote Inc.). Mice were placed in a supine position on a heated surgical platform (Indus Instruments) maintained at a temperature of 37°C and excessive hair was removed with depilatory cream (NairTM). The short-axis images were obtained of the left ventricle (LV) in B and M modes at the papillary muscle level. The systolic function of LV was evaluated by measuring left ventricular ejection fraction (LVEF). The formula to calculate LVEF was: LVEF(%) = [(LVEDV-LVESV)/LVEDV] x 100, where LVEDV and LVESV = LV end-diastolic/systolic volume [21].

## Data and statistical analysis

The parasitemia and ECG parameters from each time point were analyzed by Student's t-test and survival curves with the Mantel-Cox Long-rank test. A *P*-value less than 0.05 was considered significant statistically. The ECG parameters were also integrated into a multivariate analysis, and we performed Linear Discriminant Analysis (LDA) to evaluate the effect of *T. cruzi* infection on ECG patterns. The statistical significance of differences was assessed by Permutational Multivariate Analysis of Variance (PERMANOVA) and Bonferroni correction to adjust *P* values for multiple pairwise comparations. We further combined ECG and ECHO parameters into an LDA analysis of cardiac function.

# Results

## Susceptibility of mice to *T. cruzi* infection

To follow the progression of the *T. cruzi* infection towards the chronic phase, the blood parasite burden and survival were measured in both BALB/c and ICR mice infected with 500 blood trypomastigotes from a considered lethal dose of infection with the *T. cruzi* H1 strain. At 23

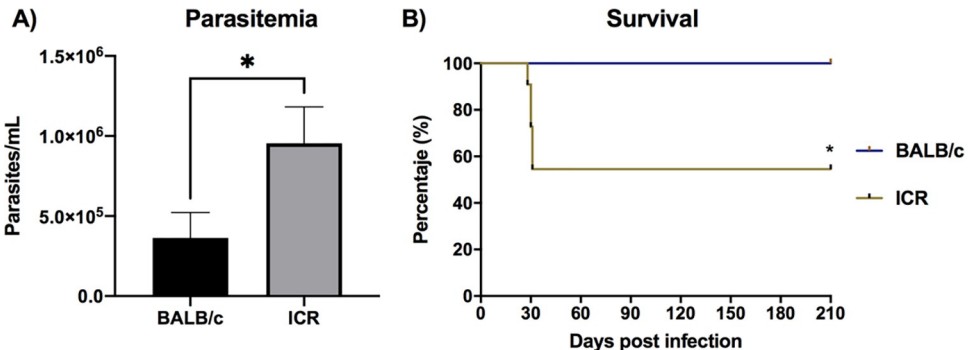

**Fig 1. Parasitemia and survival in *T. cruzi*-infected mice.** BALB/c (n = 13) and ICR (n = 13) mice were infected with 500 blood trypomastigotes of H1 *T. cruzi* by intraperitoneal injection. A total of 3 mice were used as the non-infected control group for each mouse strain and received only saline solution. **A)** Parasitemia is shown as mean ± S.D. Significant differences were calculated with the Mann-Whitney´s U-test, and is indicated as follow *, $P \leq 0.05$. **B)** Survival was monitored during 210 dpi. Significant differences were calculated using Mantel-Cox log-rank test, *, $P < 0.05$.

dpi, we observed that the levels of parasitemia were significantly higher in ICR compared to BALB/c mice (**Fig 1A**). This data was correlated with survival, as ICR mice showed high mortality 5/13 (38.47%) starting at 27 dpi and ending at day 32, while in BALB/c strain there was no mortality. Thus, all BALB/c mice survived up to 210 dpi and only 8/13 (61.53%) of ICR mice survived infection (**Fig 1B**). As expected, no mortalities were observed in non-infected control groups during that time. None mice showed sign of endpoint criteria.

## Progression of ECG alterations in *T. cruzi* infected mice

Electrical cardiac activity in non-anesthetized ICR and BALB/c mice showed differences in the infection by *T. cruzi* (from 0 to 210 dpi.). Among the significant cardiac alterations found in infected ICR mice, we observed that 100% (8/8) of mice at 37 dpi showed a decreased HR, which was correlated with an increase in the ST segment (**Fig 2A and 2G**). In addition, at the beginning of the chronic phase at 70 dpi, 87.5% (7/8) of infected ICR mice showed a decreased HR and by 105 dpi PQ and PR intervals as well as ST segment were increased compared to non-infected ICR mice (**Fig 2C, 2E and 2G**). In the late chronic phase, 87.5% (7/8) of infected ICR mice also showed an increased PQ and PR intervals at 175 dpi (**Fig 2C and 2E**) and an increased ST segment at 210 dpi compared to non-infected mice (**Fig 2G**). All this data suggests that *T. cruzi* H1 strain changes the cardiac electrical activity in ICR mice throughout infection. In contrast, no significant difference in any of the ECG parameters measured was detected in infected BALB/c mice compared to those uninfected during *T. cruzi* infection (**Fig 2B, 2D, 2F and 2H**).

## Multivariate analysis of ECG parameters during *T. cruzi* infection

We then integrated ECG parameters into a multivariate analysis to assess global changes in ECG patterns in response to *T. cruzi* infection during the acute phase of infection (0, 37 and 70 dpi) compared to the early (105 and 140 dpi) and late chronic phases (175 and 210 dpi) through Linear Discriminant Analysis (LDA) for both ICR and BALB/c mice. A significant effect of *T. cruzi* infection on ECG patterns was detected in both mouse strains (PERMANOVA F = 2.4, P = 0.033 for BALB/c and F = 8.23, P = 0.0003 for ICR mice, respectively). However, the time course of cardiac alterations differed between BALB/c and ICR mice. Indeed, combined ECG alterations observed in BALB/c mice during the acute phase of *T.*

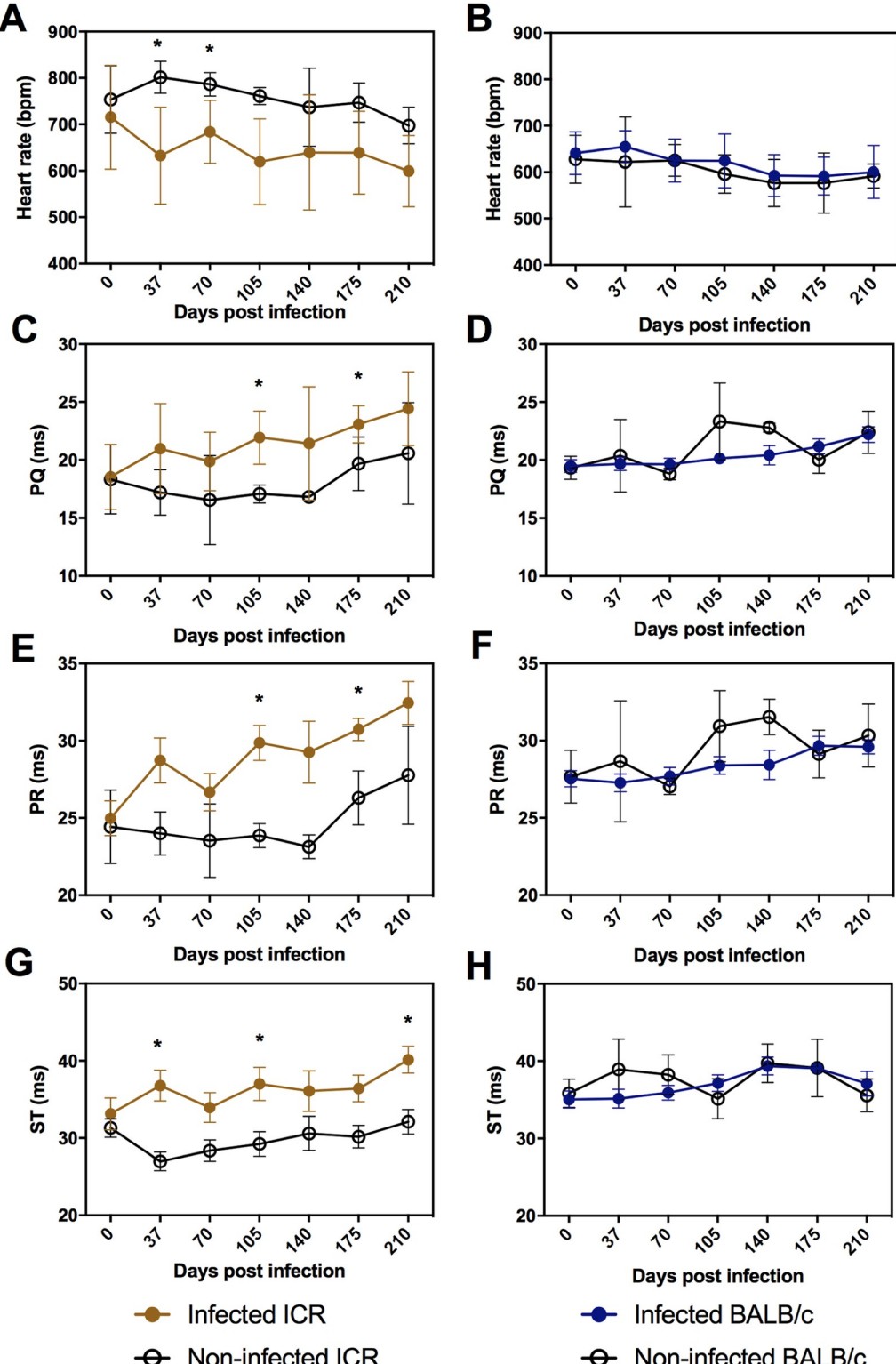

**Fig 2. ECGs from non-infected (white dots) and infected mice (marron and blue dots for ICR and BALB/c, respectively).** ECGs were recorded monthly from day 0 (before infection) to 210 dpi. The following recordings were **A, B)** Heart rate, **C, D)** PQ interval, **E, F)** PR interval and **G, H)** ST interval. Data are shown as mean ± SEM. Significant differences were estimated with Student´s t-test comparing infected and non-infected mice, and is indicated as follows *, $P \leq 0.05$.

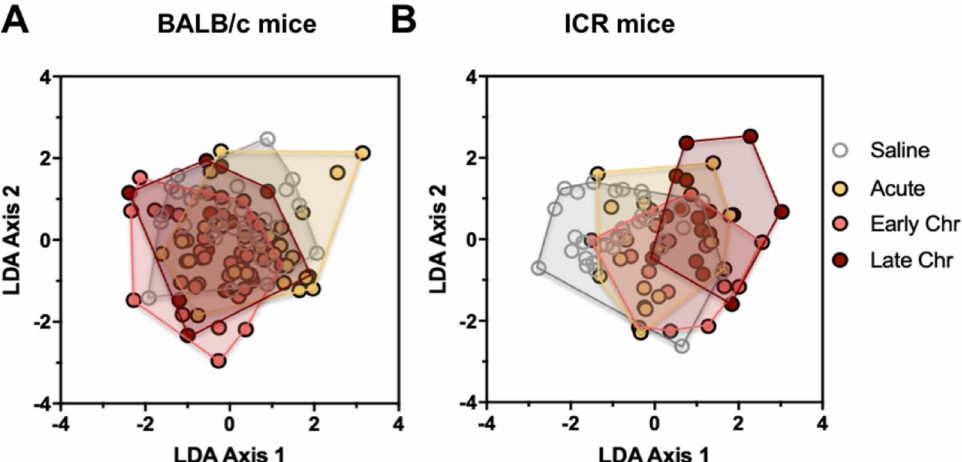

**Fig 3. Effect of *T. cruzi* infection on the ECG profiles of BALB/c and ICR mice.** ECG parameters were analyzed by LDA/PERMANOVA. **A)** BALB/c mice presented significant differences in ECG patterns following *T. cruzi* infection (PERMANOVA, *P* = 0.03), mostly during the acute phase of infection (*P* = 0.021). **B)** ICR mice also presented significant differences in ECG patterns following *T. cruzi* infection (PERMANOVA, *P* = 0.0003), for all periods (*P* = 0.031, *P* = 0.003 and *P* = 0.0006 for acute, early, and late chronic phases vs uninfected saline control, respectively. Chr: Chronic.

*cruzi* infection were contrasting compared to the chronic late phase (**Fig 3A** and **Table 1**). Conversely, ICR mice showed significant ECG alterations in all stages of infection (acute, early, and late chronic) with increased deviation from normal ECG patterns from uninfected mice over time (**Fig 3B** and **Table 1**). Accordingly, the reclassification of individual mice into their respective groups based on their ECG patterns was correct for only 45.9% of BALB/c mice, although it reached 61.5% for mice in the acute phase of *T. cruzi* infection. Reclassification was more accurate for ICR mice, reaching an average of 60.5% correct reclassification, and uninfected and late chronically infected mice were best identified with 76.4% and 75% correct reclassification, respectively (**S1 Table**). We then incorporated ECHO data from 210 dpi for the analysis of cardiac alterations in chronically infected mice. As shown in **Fig 4**, there was a major effect caused by *T. cruzi* infection on cardiac function from ICR compared to BALB/c mice in the late chronic phase. Overall, 87.5% of mice could be correctly reclassified according to their combined ECHO and ECG parameters (**S2 Table**).

## Discussion

As Chagas disease remains a public health problem, so has the milestone of developing a vaccine to halt or delay the pathology by *T. cruzi* infection [14, 16–18, 22]. Thus, an adequate murine model plays a key for the discovery and testing of new antigens in experimental

**Table 1. Pairwise comparisons of ECG patterns in *T. cruzi* infected mice.** Bonferroni-adjusted *P* values for pairwise comparisons are indicated for BALB/c mice (Above the diagonal) and for ICR mice (Below the diagonal).

|  | Uninfected | Acute | Early Chronic | Late Chronic |
|---|---|---|---|---|
| **Uninfected** | - | 0.95 | 1 | 0.92 |
| **Acute** | 0.031* | - | 0.27 | 0.021* |
| **Early Chronic** | 0.003* | 1 | - | 1 |
| **Late Chronic** | 0.0006* | 0.97 | 1 | - |

* Indicates a statistically significant difference.

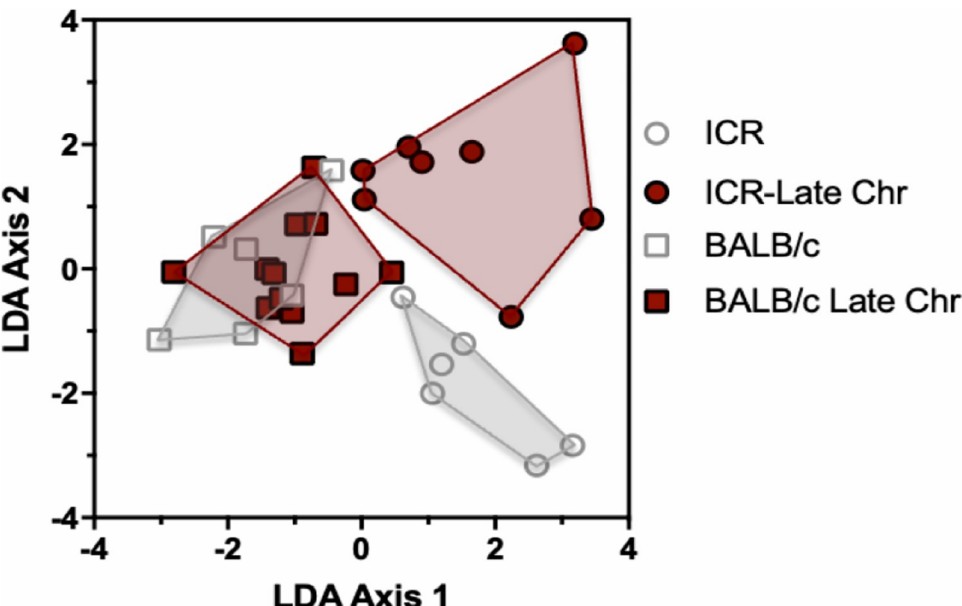

**Fig 4. Effect of *T. cruzi* infection on ECG and ECHO profiles of BALB/c and ICR mice during the late chronic phase.** Data were analyzed by LDA/PERMANOVA. Significant ECG and ECHO alterations for ICR mice at 210 days post-infection (PERMANOVA, $P = 0.009$) indicated more severe cardiac disease in ICR mice compared to BALB/c mice.

studies. Previously, BALB/c and ICR mice have been used for the study of both acute and chronic phases of experimental *T. cruzi* infection [23–26]; however, there are scarce studies focused on cardiac alterations in chronically *T. cruzi*-infected mice. Hence, in this study, we report for the first time a multi-variate analysis of the comprehensive cardiac pathology parameters in *T. cruzi*-infected BALB/c and ICR mice, indicating major differences in the progression and severity of cardiac pathology depending on the mouse strain. Our results could contribute to the selection of an adequate murine model for the study of new immunotherapies against chronic Chagas disease using a lethal/sub-lethal infective-dose for BALB/c and ICR mice by inoculation of 500 blood trypomastigotes of *T. cruzi* H1 strain, according to previous studies [20]. Here, we showed that both ICR and BALB/c mouse strains are good experimental chronic *T. cruzi*-infection models. Our data indicate that ICR mouse strain was more susceptible to *T. cruzi* H1 infection, showing a higher blood parasite burden, decreased survival and more frequent cardiac alterations compared to BALB/c mice infected with the same parasite dose.

In this study, we assessed the cardiac electrophysiology of unanesthetized mice during experimental *T. cruzi* H1 infection using electrocardiograms. It was essential to evaluate the cardiac function of unanesthetized animals, as the administration of anesthetics has to decrease the LV systolic function, HR, as well as blood pressure depending on the type of anesthetic used [10, 27, 28]. ECGs recordings showed that during *T. cruzi* infection, BALB/c mice did not exhibit electrical alterations, conversely to ICR mice which showed altered electrocardiographic parameters for all stages of infection. Upon the early stages of chronic *T. cruzi* infection, ICR mice exhibited a reduction in HR, suggestive of sinus bradycardia, at 37 and 70 dpi. However, from 105 dpi onward, infected mice demonstrated normalization of HR, indicating a potential compensatory mechanism to preserve cardiac function. At the early (105 dpi) and late (175 dpi) chronic phase, prolonged PQ and PR intervals associated with first-degree atrioventricular (AV) blockade were observed in infected ICR mice but not in BALB/c mice. In humans, Chagas disease patients with positive serology from rural communities in

Brazil and Bolivia also showed longer PQ and PR intervals, and as a consequence of these conduction disturbances patients develop first- and second-degree AV blocks [29]. The first and second-degree AV blocks are abnormalities found in *T. cruzi*-infected dogs during the acute or chronic phase [30, 31]. Likewise, first-degree AV conduction disturbance, systolic dysfunction, and ventricular arrhythmias are also ECG abnormalities found in rhesus macaques (*Macaca mulatta*) chronically infected with *T. cruzi* Colombian strain. The electrical disturbances observed in Chagas disease patients as right bundle branch block, ventricular tachycardia, second or third-degree AV block, left bundle branch block, and atrial fibrillation [32–34] are consistent with our findings in ICR mice that appear to be developing chronic chagasic cardiomyopathy. Myocardial ischemic is also a common anomaly detected in chronic Chagas disease patients with heart disease, as well as in naturally infected dogs during early and late stages of infection, with most presenting a longer ST segment [35–37]. Here, infected ICR mice also showed prolonged ST segment, suggesting myocardial ischemia in acute, early, and late chronic infection, but additionally, sinus bradycardia in the acute infection and AV blockade in early and late chronic phases whereas ECGs of BALB/c mice revealed no changes in conduction abnormalities during all stages of *T. cruzi* infection.

ECHOs further revealed a more severe dysfunction in the ejection fraction from ICR mice compared to BALB/c in the late chronic phase at 210 dpi. Consistent with our data, prior investigations of cardiac function in BALB/c mice infected with *T. cruzi* H1 strain revealed a steady ejection fraction at 208 dpi of 54.76% ± 9.35 compared to naïve mice with 48.33% ± 9 [38]. Similarly, *T. cruzi*-infected BALB/c mice exhibited an ejection fraction during chronic infection at 212 dpi with no significant differences with control mice [39].

Hence, we speculate that cardiac alterations evaluated by ECGs and ECHOs may be a result of cardiac fibrosis and inflammation caused by *T. cruzi* H1 infection. Additionally, a previous study using Brazil, Tulahuen, or Sylvio-X10/4 *T. cruzi* strains suggested that ECGs can be used as a non-invasive method to screen chronic histopathology damage in infected BALB/c mice [8].

In the last years, LDA/PERMANOVA analysis has been used as a robust method to integrate multiple variables to evaluate associations in many biological, ecological, and environmental data sets [40, 41]. In ECG data analysis, the use of univariate statistics was unable to identify significant differences in any of these parameters, hence, multivariate analysis had a greater power to identify statistical changes [42]. Our analysis allowed us to estimate the effect of *T. cruzi* infection on ECGs profiles in both murine models.

## Strengths and limitations

To our knowledge, this is the first study comparing the progression of chronic Chagas disease cardiomyopathy based on ECG and ECHO findings in two experimental *T. cruzi* (H1)-infection models. A strength of this study is that our findings align well with the three phases of *T. cruzi* infection in both murine models: an acute phase (0–70 dpi), an early chronic phase (105–140 dpi), and a late chronic phase (175–210 dpi) and shed light on the progression of cardiac disease in these two mouse strains.

A limitation of this study is that we could not measure fibrosis and inflammatory cell infiltration in the cardiac tissue of the infected mice. However, a previous study from our group showed cardiac fibrosis and inflammation after 200 dpi in BALB/c mice infected with the *T. cruzi* H1 strain [38, 39].

## Conclusion

In sum, we demonstrated that the ICR mouse strain is significantly more susceptible to *T. cruzi* H1 infection than BALB/c mice, and it develops more severe cardiac disease. This study

supports the use of both mouse strains as suitable experimental models to study chronic *T. cruzi* infection and novel treatments, as each strain presents different profiles of cardiac alterations that are consistent with the well-characterized variability observed in human Chagas disease cardiomyopathy.

## Supporting information

**S1 Table. Reclassification matrix of individual mice according to LDA analysis of ECG patterns.**
(DOCX)

**S2 Table. Reclassification matrix of individual mice according to LDA analysis of ECG patterns.**
(DOCX)

## Author Contributions

**Conceptualization:** Bárbara Carolina Arias-Argáez, Ana Paulina Haro-Álvarez, Miguel Enrique Rosado-Vallado, Julio Vladimir Cruz-Chan, Eric Dumonteil.

**Data curation:** Bárbara Carolina Arias-Argáez, Julio Vladimir Cruz-Chan, Eric Dumonteil.

**Formal analysis:** Bárbara Carolina Arias-Argáez, Julio Vladimir Cruz-Chan, Eric Dumonteil.

**Funding acquisition:** Ana Paulina Haro-Álvarez, Miguel Enrique Rosado-Vallado, Julio Vladimir Cruz-Chan, Eric Dumonteil.

**Investigation:** Victor Manuel Dzul-Huchim, Liliana Villanueva-Lizama.

**Methodology:** Bárbara Carolina Arias-Argáez, Ana Paulina Haro-Álvarez, Miguel Enrique Rosado-Vallado, Julio Vladimir Cruz-Chan, Eric Dumonteil.

**Resources:** Victor Manuel Dzul-Huchim, Liliana Villanueva-Lizama.

**Supervision:** Ana Paulina Haro-Álvarez, Miguel Enrique Rosado-Vallado, Julio Vladimir Cruz-Chan, Eric Dumonteil.

**Visualization:** Bárbara Carolina Arias-Argáez, Ana Paulina Haro-Álvarez, Miguel Enrique Rosado-Vallado, Julio Vladimir Cruz-Chan, Eric Dumonteil.

**Writing – original draft:** Bárbara Carolina Arias-Argáez, Victor Manuel Dzul-Huchim.

**Writing – review & editing:** Julio Vladimir Cruz-Chan, Eric Dumonteil.

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
