## [Decision Letter · Decision Letter 0]

13 Sep 2023

PONE-D-23-22437Signature of cardiac alterations in early and late chronic infections with Trypanosoma cruzi in micePLOS ONE

Dear Dr. Dumonteil,

Thank you for submitting your manuscript to PLOS ONE. After careful consideration, we feel that it has merit but does not fully meet PLOS ONE’s publication criteria as it currently stands. Therefore, we invite you to submit a revised version of the manuscript that addresses the points raised during the review process.

ACADEMIC EDITOR: Please revise the manuscript based on reviewers' comments. 

We look forward to receiving your revised manuscript.

Kind regards,

Amir Hossein Behnoush

Academic Editor

PLOS ONE

Journal Requirements:

"This work was supported by Fundación Carlos Slim (#87714) and Consejo Nacional de Ciencia y Tecnología, México (CONACYT, #PDCPN2015-102). B.C.A.A was supported by scholarship #334483 from CONACYT."

Reviewers' comments:

Reviewer's Responses to Questions

**Comments to the Author**

1. Is the manuscript technically sound, and do the data support the conclusions?

Reviewer #1: Yes

Reviewer #2: Yes

2. Has the statistical analysis been performed appropriately and rigorously? 

Reviewer #1: Yes

Reviewer #2: Yes

3. Have the authors made all data underlying the findings in their manuscript fully available?

Reviewer #1: Yes

Reviewer #2: Yes

4. Is the manuscript presented in an intelligible fashion and written in standard English?

Reviewer #1: Yes

Reviewer #2: Yes

5. Review Comments to the Author

Reviewer #1: The manuscript entitled "Signature of cardiac alterations in early and late chronic infections with Trypanosoma cruzi in mice" evaluated cardiac alterations associated with Chagas disease. I have minor concerns:

1- Add the conclusion to the abstract.

2- Abbreviations should be defined in their first use. For example, "dpi" should be defined in the main text.

3- There are several typos and grammar errors in the manuscript. Revise the whole paper in this regard.

4- In the discussion section, the first paragraph should mention the main findings of your study. Please add your most valuable and novel findings of your paper to the first paragraph.

5- Add new headings named "Strengths and limitations" and "Conclusions" to the discussion section. Add the strengths of your study.

Reviewer #2: I consider that the study is , simple, and clear but very interesting. The conclusions are in accordance with the results obtained . To enhance the presentation of the manuscript, I have contributed with some corrections of form with the correct names of the mice strains used and the forms of expression of some numbers or decimal or signs in papers.

Also, i have modified some mistakes and the way in which we have to call the patients

6. PLOS authors have the option to publish the peer review history of their article (what does this mean?). If published, this will include your full peer review and any attached files.

Reviewer #1: No

Reviewer #2: No

---

## [Author Response · Author response to Decision Letter 0]

21 Sep 2023

Response to reviewers

Reviewer #1: The manuscript entitled "Signature of cardiac alterations in early and late chronic infections with Trypanosoma cruzi in mice" evaluated cardiac alterations associated with Chagas disease. I have minor concerns:

1- Add the conclusion to the abstract.

RESPONSE: We thank the reviewer 1 for improving the MS. We have added the conclusion in the Abstract section. Please see lines 46-50 in Tracked version.

2- Abbreviations should be defined in their first use. For example, "dpi" should be defined in the main text.

RESPONSE: We have revised all the MS correctly writing the abbreviations.

3- There are several typos and grammar errors in the manuscript. Revise the whole paper in this regard.

RESPONSE: We have revised all the MS to avoid typos and grammar errors. We thank reviewer 1 for noticing the spelling errors in the MS

4- In the discussion section, the first paragraph should mention the main findings of your study. Please add your most valuable and novel findings of your paper to the first paragraph.

RESPONSE: We appreciate the reviewer's comment. We have rewritten the first paragraph of the Discussion section. Please see lines 275-279 in Tracked version.

5- Add new headings named "Strengths and limitations" and "Conclusions" to the discussion section. Add the strengths of your study.

RESPONSE: We have added "Limitations and strengths" and "Conclusions" sections in the MS. Please see lines 347-366 in Tracked version.

Reviewer #2: I consider that the study is, simple, and clear but very interesting. The conclusions are in accordance with the results obtained. To enhance the presentation of the manuscript, I have contributed with some corrections of form with the correct names of the mice strains used and the forms of expression of some numbers or decimal or signs in papers.

Also, I have modified some mistakes and the way in which we have to call the patients

RESPONSE: We appreciate all the comments recommended by the reviewer #2 to improve the manuscript. We have revised cautiously the MS, replacing words as” Balb/c” and “chagasic”. We have added a space between the last number and the % sign in all the MS. Italics words have been added appropriately.

---

## [Editor Report · Decision Letter 1]

25 Sep 2023

Signature of cardiac alterations in early and late chronic infections with Trypanosoma cruzi in mice

PONE-D-23-22437R1

Dear Dr. Dumonteil,

We’re pleased to inform you that your manuscript has been judged scientifically suitable for publication and will be formally accepted for publication once it meets all outstanding technical requirements.

Kind regards,

Amir Hossein Behnoush

Academic Editor

PLOS ONE
---

## [Editor Report · Acceptance letter]

28 Sep 2023

PONE-D-23-22437R1 

Signature of cardiac alterations in early and late chronic infections with *Trypanosoma cruzi* in mice 

Dear Dr. Dumonteil:

I'm pleased to inform you that your manuscript has been deemed suitable for publication in PLOS ONE. Congratulations! Your manuscript is now with our production department. 

Kind regards, 

on behalf of

Dr. Amir Hossein Behnoush 

Academic Editor

PLOS ONE